Investigating the regulatory role of HvANT2 in anthocyanin biosynthesis through protein–motif interaction in Qingke

Wang Yan 1 2 3 4
Chen Lin 1 2 3 4
Yao Youhua 1 2 3 4
Chen Lupeng 1 2 3 4
Cui Yongmei 1 2 3 4
An Likun 1 2 3 4
Li Xin 1 2 3 4
Bai Yixiong 1 2 3 4
Yao Xiaohua yaoxiaohua009@126.com 1 2 3 4
Wu Kunlun wklqaaf@163.com 1 2 3 4
1 Academy of Agricultural and Forestry Sciences, Qinghai University , Xining , Qinghai , China
2 Laboratory for Research and Utilization of Qinghai Tibet Plateau Germplasm Resources , Xining , Qinghai , China
3 Qinghai Key Laboratory of Hulless Barley Genetics and Breeding , Xining , Qinghai , China
4 Qinghai Subcenter of National Hulless Barley Improvement , Xining , Qinghai , China
Orlov Yuriy
Electronic publication date: 2024 Jul 10
Publication date: 2024
Volume: 12
Electronic Location ID: e17736
Received 2024 Feb 12; Accepted 2024 Jun 23
Copyright: ©2024 Wang et al.
Copyright year: 2024
Copyright holder: Wang et al.
License: This is an open access article distributed under the terms of the Creative Commons Attribution License, which permits unrestricted use, distribution, reproduction and adaptation in any medium and for any purpose provided that it is properly attributed. For attribution, the original author(s), title, publication source (PeerJ) and either DOI or URL of the article must be cited.
License URL: https://creativecommons.org/licenses/by/4.0/

Keywords: Qingke, Anthocyanin, HvANT2, HvbHLH gene family, Y1H

Funding: The Natural Science Foundation of China 31960427 The National Key Research and Development Project 2022YFD2301300 The Construction Project for Innovation Platform of Qinghai Province 2023_1_5 The Agriculture Research System of China CARS-05 This research was supported by the Natural Science Foundation of China (31960427), the National Key Research and Development Project (2022YFD2301300), the Construction Project for Innovation Platform of Qinghai Province (2023_1_5), and the Agriculture Research System of China (CARS-05). The funders had no role in study design, data collection and analysis, decision to publish, or preparation of the manuscript.

==============================
Background

Currently, there are no reports on the HvbHLH gene family in the recent barley genome (Morex_V3). Furthermore, the structural genes related to anthocyanin synthesis that interact with HvANT2 have yet to be fully identified.

Methods

In this study, a bioinformatics approach was used to systematically analyze the HvbHLH gene family. The expression of this gene family was analyzed through RNA sequencing (RNA-seq), and the gene with the most significant expression level, HvANT2, was analyzed using quantitative reverse transcription polymerase chain reaction (qRT-PCR) in different tissues of two differently colored varieties. Finally, structural genes related to anthocyanin synthesis and their interactions with HvANT2 were verified using a yeast one-hybrid (Y1H) assay.

Results

The study identified 161 bHLH genes, designated as HvbHLH1 to HvbHLH161, from the most recent barley genome available. Evolutionary tree analysis categorized barley bHLH TFs into 21 subfamilies, demonstrating a pronounced similarity to rice and maize. Through RNA-Seq analysis of purple and white grain Qingke, we discovered a significant transcription factor (TF), HvANT2 (HvbHLH78), associated with anthocyanin biosynthesis. Subsequently, HvANT2 protein-motifs interaction assays revealed 41 interacting motifs, three of which were validated through Y1H experiments. These validated motifs were found in the promoter regions of key structural genes (CHI, F3’H, and GT) integral to the anthocyanin synthesis pathway. These findings provide substantial evidence for the pivotal role of HvANT2 TF in anthocyanin biosynthesis.

Introduction

The basic Helix-Loop-Helix (bHLH) family is widely distributed among plants, animals, and microorganisms, exerting pivotal regulatory functions in plant growth and development (Zuo, Lee & Kang, 2023; Jia et al., 2023). These protein sequences feature a highly conserved bHLH domain that typically spans 50–60 amino acids (Ye et al., 2021; Hao et al., 2021). The first bHLH transcription factor (TF) was identified in mice (Heim et al., 2003), while the Lc protein encoded by the R gene in maize, which is primarily involved in anthocyanin biosynthesis, was the first bHLH TF identified in plants (Ludwig et al., 1989). Numerous bHLH TFs have since been subsequently identified and isolated in various plant species, with the entire bHLH family being analyzed in plants, including in Arabidopsis thaliana, Zea mays, Aquilaria sinensis, and Dendrobium officinale (Hao et al., 2021; Zhang et al., 2018; Sun et al., 2022; Wang & Liu, 2020; Li et al., 2006). Functional studies of plant bHLH family proteins have become a prominent and rapidly developing field in recent years. These TFs have actively contributed to the development of diverse aspects of plant biology, including plant epidermal and root hair development, photomorphogenesis, light signaling, and overall tissue and organ growth (Zuo, Lee & Kang, 2023; Wu et al., 2021; Ren et al., 2021). Moreover, the bHLH TF family plays a pivotal role in enhancing plant resistance to harsh environmental stressors such as drought, salt, and cold (Ndayambaza et al., 2021; Fan et al., 2021).

Qingke (Hordeum vulgare L.var. nudum Hook. f), also known as naked barley, is a member of the Barley genus within the Gramineae family. It is well-suited for cultivation in highland regions due to its short reproductive period, high cold resistance, and wide adaptability (Guo et al., 2020; Wang et al., 2022). Recent scientific research and industrial advancements have uncovered a myriad of nutritional and health benefits associated with Qingke, accentuating its versatility. Qingke has long been a staple food for Tibetans, boasting a richness in β-glucans and other functional constituents, making it a high-quality and healthy ingredient (Lin et al., 2018). Furthermore, Qingke is renowned for its abundant protein content, many vitamins, and low-fat levels, enabling it to reduce blood lipids, cholesterol, and blood sugar, as well as to prevent obesity (Guo et al., 2020; Guo et al., 2018; Hong et al., 2023). Notably, colored varieties of Qingke exhibit superior nutrient profiles compared to their regular counterparts, especially in terms of anthocyanin diversity and content (Dang et al., 2022). Anthocyanin, the largest group of secondary plant metabolites among flavonoids, possesses antioxidant effects and can be used as a natural antioxidant (Ma et al., 2021; Sunil & Shetty, 2022). As a result of its rich nutritional content and development potential, Qingke, the staple food of many Tibetan communities, holds great value in the food industry, particularly in the development of colored highland barley resources.

This study utilized the most recent genome-wide data of barley, focusing on Qingke, as it is considered a subspecies of barley. Initially, the chromosomal localization, gene and protein structure, phylogeny, and intergenic regulation of bHLH TFs binding to motifs of 161 genes from the bHLH TF family were studied, and they were subsequently compared to other species such as Arabidopsis thaliana, Oryza sativa, and Zea mays. It is worth noting that the entire bHLH gene family in barley has previously been mapped twice, but both of these studies relied on an outdated reference genome (Quan et al., 2023; Ke et al., 2020). Furthermore, discrepancies were observed in the number of genes identified, with a variation of up to 38 genes. As a result, our re-identification was essential to ensure the accuracy and comprehensiveness of understanding the bHLH gene family in barley. Additionally, we analyzed the expression of 161 HvbHLH TFs using RNA-seq data from Qingke seeds with varying grain colors that had been previously collected by our research team. HvANT2 was identified as the candidate gene associated with anthocyanin synthesis in purple seeds, which was consistent with the results of previous studies indicating that ANT2 was the primary gene associated with anthocyanin synthesis (Cockram et al., 2010). The yeast one-hybrid (Y1H) study on the HvANT2 gene of purple grains facilitated the initial construction of a gene regulatory network of HvANT2 linked to Qingke purple grains. These results will provide a basis for further research on the HvbHLH family.

Materials & Methods

Plant materials

The purple barley variety Nierumuzha and the white barley variety Kunlun 10 were procured from the College of Agriculture and Forestry at Qinghai University (Xining, Qinghai, China). Seeds from both varieties were harvested at different growth stages following Zadoks’ growth scale (Zadok, Chang & Konzak, 1974), including the early milk ripening, late milk ripening, and soft dough stages, for subsequent RNA-seq analysis.

Gene identification

The bHLH TF families were retrieved from the barley genome (http://doi.org/10.5447/ipk/2021/3). A comprehensive genome-wide search (E-value cutoff = 0.0001 threshold filter) was executed using HMMER 3.2.1 to identify the bHLH gene families. Subsequently, the extracted protein sequences were manually verified to confirm the presence of conserved structural domains. This verification process involved consulting the SMART database to cross-reference the obtained homologous family genes.

Gene location mapping

The positional information of all bHLH TFs in the barley genome was extracted, and the chromosomal locations of barley bHLH family genes were visualized and mapped using TBtools (version 1.12).

Structure and motif analysis of genes

The online MEME web server (https://meme-suite.org/meme/tools/meme) was utilized to search for conserved motifs within the barley bHLH protein. The search criteria involved motifs with lengths ranging from 6 to 50, with an optimal length of 10, and with other parameters set to the system defaults. Following motif discovery, the relevant masthead files were downloaded, and the gene structure and motif visualization of the bHLH gene family members were performed using the gene structure view program available in TBtools.

Analysis of covariance between species of the bHLH gene family

Genome-wide covariance analyses of barley with Arabidopsis thaliana, Oryza sativa, and Zea mays were carried out using TBtools “One Step MCScanX” (Threshold: E-value = 1e−10, Num of Blast Hits = 5), and the results of inter-genome gene pairings were obtained. Subsequently, the gene pairing information of the gene families was extracted by using in-home Perl scripts, and finally, covariance maps were plotted using TBtools. BLASTp and MCScanX were used to analyze gene duplication events.

Multi-species evolutionary tree construction

To conduct sequence alignment, the bHLH TFs of barley and the bHLH family genes of Arabidopsis were downloaded from the http://planttfdb.gao-lab.org/index.php database. These sequences were subjected to alignment using MUSCLE software (v3.8.31), followed by further processing and alignment using TBtools. Subsequently, a phylogenetic tree was analyzed with FastTree (FastTree 2.1), after which it was visualized and drawn using iTOL (https://itol.embl.de).

Gene expression analysis in RNA-seq data

Utilizing RNA-seq data from our prior research on Qingke with different grain colors (Yao et al., 2022), we retrieved the number of enzyme fragments per million (FPKM) values for the bHLH family at each developmental stage using the OmicStudio platform. The obtained expression data were collated and subjected to analysis using Microsoft Excel 2010 and SPSS 22.0 statistical software.

Analysis of the expression pattern of the HvANT2 gene

Quantitative primers were designed using Primer 5.0. The upstream primer sequence was GAAGAAAGCTTTGGCCGGTG, and the downstream primer sequence was TCCACCTTGTGAATGGACGG. cDNAs from tissues (roots, stems, leaves, awns, and seed coats) of Nierumuzha and Kunlun 10 cDNAs at three different stages (early milk, late milk and early dough) after sowing were used as templates, and 18SrRNA was selected as an internal reference gene. TaKaRa’s TB GreenpremixExTaq II fluorescent dye was used for RT-qPCR, which was conducted on a LightCycler 480 system. The qRT-PCR reaction procedure and system setup were conducted with reference to the method of Yao et al. (2022). To calculate the relative expression of genes in different stages for Qingke, the 2−ΔΔCt formula was applied (Yao et al., 2022). It is worth noting that all experiments were conducted with three biological replicates, and the experimental data were statistically analyzed through ANOVA using SPSS 22.0.

Identification of motifs that interact with the transcription factor HvANT2

To identify motifs interacting with HvANT2, the HvANT2 gene itself, constructed on the pGADT7 vector (ProNet Biotech Co., Ltd., Nanjing, China), was used as bait to screen a Y1H motif library (ProNet Biotech Co., Ltd., Nanjing, China). Motifs interacting with pGADT7-HvANT2 were identified by DNA sequencing and comparative analysis of positive clones. The sequences of structural genes containing these motifs (Table S1) were obtained from the online website Ensembl Plants (https://plants.ensembl.org/index.html).

Analysis of DNA and protein interaction

Based on the motifs that interacted with HvANT2, three key structural genes (CHI, F3′H, and GT) involved in the anthocyanin synthesis pathway were selected (Yao et al., 2022). The full-length CDS of HvANT2 was inserted into pGADT7 (ProNet Biotech Co., Ltd., Nanjing, China) to generate the AD-HvANT2 structure. The promoter fragment of these selected genes were cloned into the pHIS2 vector (ProNet Biotech Co., Ltd., Nanjing, China) to generate pHIS2- (pHIS2-CHI, pHIS2- F3′H, and pHIS2-GT), and it was then transformed into the Y1HGold yeast strain (ProNet Biotech Co., Ltd., Nanjing, China). The yeast colonies were subsequently transferred to plates containing SD/-Trp/-Leu and SD/-His/-Trp/-Leu media supplemented with 50 mM 3-AT and allowed to grow at 30 °C for 3–5 days.

Results

Chromosome location of HvbHLH genes

The barley genome and proteome databases were searched using the HMM model to screen and identify genes encoding the barley HvbHLH proteins. After eliminating redundant proteins, a total of 161 HvbHLH TFs were identified (Table S2). To illustrate the chromosomal distribution of the HvbHLH TFs, their locations were mapped based on their starting positions on the chromosomes (Fig. 1). It was found that 161 barley HvbHLH TFs were successfully localized on the seven chromosomes of barley, displaying a heterogeneous distribution of HvbHLH TFs. Notably, chromosome 7 exhibited the highest number, with 31 HvbHLH TFs, while chromosome 1 contained the fewest, with only 12 HvbHLH TFs.

Phylogenetic analysis of the HvbHLH protein family

An unrooted phylogenetic tree was constructed using full-length amino acid sequences to analyze the evolutionary relationships among the bHLH TFs in Arabidopsis and barley (Fig. 2). The bHLH family was divided into 21 distinct groups, denoted as A-U. The R subfamily had the largest number of bHLH members, comprising 29 bHLH units, while the B subfamily had the lowest number, with only four members. Moreover, subfamilies B, C, and L exclusively contained genes from barley. The phylogenetic tree of barley HvbHLH TFs revealed that proteins with similar sequence lengths and structural domain positions formed distinct clusters.

Analysis of HvbHLH gene structure and protein conserved motifs

We examined the intron structure to gain insight into the gene structure of HvbHLH TFs. The analysis revealed that the intron numbers ranged from 1 to 12. Specifically, HvbHLH57 (HORVU.MOREX.r3.5HG0488610) had the highest count of 12 (Fig. 3C). Moreover, HvbHLH from the same subfamily showed similarities in their intron structures.

To elucidate the structural diversity of bHLH proteins in barley, the 161 bHLH proteins of barley were subjected to evolutionary tree construction (Fig. 3A) and conserved motif analysis (Fig. 3B) using TBtools software, which led to the identification of 10 conserved motifs (Fig. S1). The results showed that members of the bHLH TF family within the same subfamily exhibited similar types and quantities of motifs; despite this, variations in motif patterns were observed among members of the same subfamily. For example, subfamilies III, IV, and VI all have the same motifs and are aligned. Subfamily I has motifs that are largely similar, but some motifs are contiguous while others are separated. Overall, barley bHLH proteins typically displayed relatively simple motif configurations, with most sub-families comprising 2–3 members each. Notably, HvbHLH54 (HORVU.MOREX.r3.4HG0362890), HvbHLH92 (HORVU.MOREX.r3.2HG0110130), HvbHLH115 (HORVU.MOREX.r3.7HG0750050), and HvbHLH157 (HORVU.MOREX.r3.7HG0642120) had the simplest motif configurations, each with only one motif. Conversely, HvbHLH15 (HORVU.MOREX.r3.5HG0493040) and HvbHLH147 (HORVU.MOREX.r3.7HG0703220) had the most abundant motifs, with each having five distinct motifs.

Figure 1 Chromosome distribution of HvbHLHs.

(A) Chromosomal distribution of HvbHLH TFs. (B) Pie chart showing the number of HvbHLH TFs distributed on chromosomes.

Figure 2 Phylogenetic tree of bHLH TF in Arabidopsis and barley.

All bHLH domains are clustered into 21 sub-branches (denoted by letters A–U).

Figure 3 Conserved motif and gene structure analysis of the barley bHLH gene family.

(A) Phylogenetic tree of barley bHLH gene neighbor-joining using the maximum likelihood method and 1,000 bootstrap replicates. (NJ). (B) Conserved motif analysis of barley bHLH gene. Sequence information for each motif is provided in Fig. S1. (C) Gene structure analysis of the barley bHLH gene, including UTR, CDS, and introns. Yellow boxes indicate CDS; green boxes indicate UTR and black lines indicate introns.

Furthermore, certain motifs were specific to particular subfamilies, such as motif nine, which exclusively appeared within a single subfamily, functioning as a hallmark conserved motif for that subfamily. Similarly, Pattern 6 is only found in Subfamily V. Motif 3 was found in 38 HvbHLH proteins, predominantly originating from evolutionary branches I, II, III, IV, V, VI, and VII. Motif 4 was found in 12 HvbHLH proteins (HvbHLH12, 44, 91, 115, 81, 73, 39, 6, 42, 3, 148, and 89). Motif 5 was found in 54 HvbHLH proteins that primarily originated from evolutionary branches X, XI, XII, XIII, XIV, XV, and XVI. Motif 7 was found in 18 HvbHLH proteins originating primarily from evolutionary branches III and VIII. Motif 8 was found in seven HvbHLH proteins (HvbHLH82, 7, 68, 77, 123, 78, and 122) that were mainly derived from evolutionary branches XV and XIII. Motif 10 was found in five HvbHLH proteins (HvbHLH15, 2, 94, 16, and 147) predominantly derived from evolutionary branches V and XIV. Additionally, we found that Motif 1 and Motif 2 were characteristic conserved motifs situated within the bHLH conserved structural domain, and almost all genes include Motif 1 and Motif 2, with only HvbHLH157 containing only Motif 1, suggesting their potential significance in influencing the functionality of barley HvbHLH TFs.

Analysis of covariance between barley and the bHLH family of Arabidopsis, rice, and maize

In this study, we analyzed the covariance of the barley bHLH TF with Arabidopsis, rice, and maize using TBtools (Fig. 4A). The results showed that the size of covariance fragments was closely related to the timing of gene evolution (McCouch, 2001). Analyses of covariance between genes of different species could inform studies of their evolutionary relationships. The results revealed that barley had 18 covariant gene pairs with Arabidopsis, 145 covariant gene pairs with rice, and 186 covariant gene pairs with maize (Table S3). These findings implied that the barley bHLH TF covaried with Arabidopsis in a shorter timeframe compared to rice and maize. Consequently, it was hypothesized that the barley bHLH TFs share a higher degree of homology with rice and maize than with Arabidopsis.

Figure 4 Analysis of covariance between barley and the bHLH family of Arabidopsis, rice, and maize.

(A) Homology analysis of bHLH TFs in barley and three plants (Arabidopsis thaliana, rice, and maize). The grey line shows the co-lined blocks in the genomes of barley and the other plants, while the blue line highlights the co-lined bHLH pairs. (B) Sequence analysis of the barley HvbHLH TFs. Brown lines indicate segmental duplication gene pairs.

Furthermore, our investigation delved into the role of segmental and tandem duplications within the HvbHLH gene family. Our findings revealed the existence of 24 homologous gene pairs among the 161 HvbHLH TFs, comprising 16 segmental duplicate pairs and eight tandem duplicate pairs (Fig. 4B). Simultaneously, we analyzed the covariance of genes within the same species. It is noteworthy that tandem and segmental duplications have played a pivotal role in the development of the gene family members and the evolution of plant genomes.

Expression profiling of the HvbHLH genes in kernels of white and purple grain colors of Qingke

Using RNA-seq data obtained from the seed coats of white-grained Qingke (Kunlun 10) and purple-grained Qingke (Nierumuzha), we analyzed the transcription expression levels within the HvbHLH gene family at different growth stages. The results showed that 142 HvbHLH genes were expressed and that 19 genes had no detectable expression in both varieties. The unexpressed genes in the seed coat of these species were hypothesized to either have no function or to be expressed in other tissues (Fig. 5A).

Figure 5 Expression profiling of HvbHLH genes in different coloured barley tissues.

(A) Heatmap of HvbHLH gene expression in the seed coat of Nierumuzha and Kunlun 10 at three stages of time (Nierumuzha: the early lactation is denoted by P1, late lactation is denoted by P2, and soft dough is denoted by P3; Kunlun 10: the early lactation is denoted by W1, W2 denotes late lactation, and W3 denotes soft dough), FPKM values of HvbHLH genes transformed by log2 and heatmap constructed by TBtools. (B) Expression pattern of HvbHLH78 (Ant2) in different tissues of Nierumuzha and Kunlun 10. A, B, C, and D indicate highly significant differences (P < 0.01). Data shown in histograms are expressed as mean ± SD. (C) Genes with highly significant differences in Nierumuzha and Kunlun 10 (P < 0.01). (Nierumuzha: the early lactation is denoted by P1, P2, and P3 denotes late lactation denotes soft dough; Kunlun 10: the early lactation is denoted by W1, W2, and W3 denote late lactation denotes soft dough). FPKM values of HvbHLH genes transformed by log2 and heatmap constructed by TBtools. The numbers in the rectangles indicate the expression in RNA-seq.

Furthermore, we identified 17 TFs that displayed significant differences between the two varieties (P < 0.01) (Fig. 5C). Notably, the TF HvbHLH78, also known as HvAnt2, exhibited minimal expression during the first three stages of white grains and the first two stages of purple grains. However, its expression significantly increased during the late milk and early dough stages of purple grains compared to each period of white grains (P < 0.01) (Fig. 5C). In the early dough stage, the expression level of HvAnt2 in purple grains was approximately 100 times higher than that in white grains. Additionally, in the late milk stage, the expression level of HvAnt2 in purple grains was approximately 40 times higher than that in white grains (Fig. 5C). This observation led to the hypothesis that the HvAnt2 TF plays a pivotal role in the regulation of grain color.

To investigate the expression pattern of the HvAnt2 gene in color formation in barley, we employed quantitative real-time polymerase chain reaction (qRT-PCR) to measure the relative expression levels of HvAnt2 in various tissues (roots, stems, leaves, awns, and seed coats) and at different stages in two different varieties. The results revealed low or almost negligible expression levels of HvAnt2 in all tissues of Kunlun 10. However, in Nierumuzha, HvAnt2 showed tissue-specific expression, with extremely low expression levels in roots, stems, leaves, and awns, but significantly high expression levels in seed coats. The expression levels in seed coats were tremendously elevated during the late milk stage and the early dough stage (P < 0.01), reaching their highest level in the late milk stage, which was approximately 16 times higher than that in the early milk stage (Fig. 5B). This result caused us to take great interest in the HvAnt2 gene.

Motifs interacting with the transcription factor HvANT2

We successfully obtained 41 motifs (Table 1). Subsequent re-verification experiments confirmed that all 41 motifs identified through the screening were able to grow on SD-TL, SD-TLH, and SD-TLH+50 mM 3AT plates (Fig. 6). Eventually, we identified three enzymes encoding anthocyanin structural genes, namely, chalcone isomerase (CHI), dihydroflavonol-3′-hydrogenase (F3′H), and UDP-glucosyltransferase (GT). These genes contain motifs that interact with HvANT2. Two motifs (TGCAAGC and TGCGGCC) were found in the promoter region of the CHI gene, two motifs (GAGGAAC and AGTGCAG) were found in the F3′H gene, and six motifs (GGTACTC, TCGGTCC, GTATCAT, TGCGGCC, ACAAAA, and AAGTACG) were found in the GT gene.

Table 1 41 motifs of different sequences.

Number	Result	Number	Result	Number	Result	Number	Result	
2	AGTGCAG	14	AGCCACC	27	CTAAAAT	38	AGAGATC	
3	GCGCAGC	15	ACCCGGT	28	ACAAAAA	39	GTTGTGG	
4	GTCGGCG	16	GAGGAAC	29	GGTACTC	40	CGGTATG	
5	GCGGACA	17	GTATCAT	31	TTAGCCC	41	GAGTGGC	
6	GTTCTTA	20	ACATCGA	32	AGTACGC	43	TCACTTA	
7	CGCTTAT	21	TGAGCTC	33	GTCTCCC	44	AGCATTC	
8	CCTAAGC	23	GTACCGG	34	TGCAAGC	45	TATCAGT	
9	GATACCT	24	GCTAAAG	35	AAGTACG	46	GACCCAA	
11	CTCTGGT	25	TAAAGAC	36	GTTTAGC	47	TGCGGCC	
12	TCGGTCC	26	AGACACA	37	CAGTGGC	48	CCGACGC	
13	TAAGAGG							

Figure 6 Slalom validation of 41 motifs with different sequences.

(+): pGAD53m+pHIS2-p53 as positive control, (-): pGADT7-ANT2+pHIS2-p53 as negative control. SD-TL: -trp, -leu; SD-LH: -leu, -his; SD-TLH: -trp, -leu, -his. 3AT: a competitive inhibitor of yeast HIS2 protein synthesis, used to inhibit leaky expression of the HIS2 gene.

Motif interactions of HvANT2 with promoters of HvCHI, HvF3′H, and HvGT

The motifs obtained from the library were screened for motifs present in the promoter regions of structural genes involved in the anthocyanin synthesis pathway. The co-transformation process involved introducing the pHIS2 vector containing the CHI, F3′H, and GT genes, along with the pGADT7 vector containing the HvANT2 protein, was performed in the yeast strain Y187. In the Y1H experiment, the combinations pHIS2-CHI + pGADT7-HvANT2, pHIS2-F3′H + pGADT7-HvANT2, and pHIS2-GT + pGADT7-HvANT2 exhibited normal growth on selective media (SD-TL, SD-TLH, and SD-TLH + 50 mM 3AT), indicating positive growth (Fig. 7). These results demonstrate that pHIS2-CHI, pHIS2-F3′H, and pHIS2-GT interact with pGADT7-HvANT2 (Fig. 8).

Figure 7 Interaction validation results of pHIS2-CHI, pHIS2-F3′H, and pHIS2-GT with pGADT7-ANT2.

(+): pGAD53m+pHIS2-p53 as positive control, (-): pGADT7-ANT2+pHIS2-p53 as negative control. SD-TL: -trp, -leu; SD-LH: -leu, -his; SD-TLH: -trp, -leu, -his. 3AT: a competitive inhibitor of yeast HIS2 protein synthesis, used to inhibit leaky expression of the HIS2 gene.

Figure 8 Network map of HvAnt2 regulation in purple-grained Qingke.

Discussion

The bHLH TF family, considered one of the largest TF families, is widely distributed in all three eukaryotic kingdoms (Carretero-Paulet et al., 2010). In recent years, there has been a growing effort to functionally characterize bHLH proteins in various plant species, resulting in the identification of numerous members, such as the 124 bHLH TFs identified in potato (Wang et al., 2018), the 230 bHLH TFs identified in Chinese cabbage (Song et al., 2014), the 225 bHLH TFs identified in wheat (Guo & Wang, 2017), and the 169 bHLH TFs identified in ginseng (Chu et al., 2018). In our current study, we identified 161 bHLH TFs in barley, which slightly exceeded the quantities reported in previous studies, which found 103 (Quan et al., 2023) and 141 (Ke et al., 2020) bHLH TFs in barley. This variation could be attributed to our use of the most recent barley reference genome data and differences in screening methodologies. Our analysis of the full-length sequences of HvbHLH proteins indicated that all of these TFs contained characteristic bHLH structural domains or that they had been appropriately annotated. Additionally, the chromosomal localization analysis demonstrated that barley bHLH TFs exhibited a clustered distribution, forming gene clusters where two or more genes were grouped (Fig. 1).

The results of the evolutionary analysis demonstrated that HvbHLH can be categorized into 21 subfamilies (Fig. 2). In plants, Pires & Dolan (2010) analyzed more than 500 sequences of bHLH proteins derived from algae and terrestrial plants and classified them into 26 subfamilies; In the case of chili (Capsicum annuum L), its 122 bHLH proteins were classified into 21 subfamilies (Zhang et al., 2020b); while in apple, its 188 bHLH proteins were classified into 18 subfamilies (Mao et al., 2017). In contrast to Arabidopsis, barley exhibited three subclasses (B, C, and L) that exclusively contained members of the HvbHLH family, suggesting that these might be the result of gene insertions that had occurred in barley during evolution. Additionally, some subclasses (e.g., A, D, and E), within the HvbHLH family contained just one member, suggesting relatively slow evolution and a potential for functional conservation, implying their importance in regulating growth and development in barley. Barley had the highest number of bHLH members in subclass H (16), suggesting that these bHLH gene clusters might have undergone stronger amplification during the long evolutionary process that is characteristic of monocotyledons.

Conservative sequence and gene structure analysis provided crucial information for resolving evolutionary relationships (Hong et al., 2019). Most HvbHLH genes within the same subfamily shared similar sequences and exon–intron arrangements (Fig. 3). The results of the analysis of HvbHLH conserved motifs revealed that Motif 1 and Motif 2 were characteristic conserved motifs (Fig. 3). Motif 1 and Motif 2 were believed to play a significant role in the functional activity of HvbHLH (Hong et al., 2019). Additionally, the composition of other sequences was unique, with motif conservation observed across subgroups. For instance, Motif 6 was exclusive to Subgroup V, while Motif 9 was present only in the XV group. These results suggested that these features not only enhance the diversity of HvbHLH proteins among different subgroups, but that they also might have specific functions. The exact functional mechanisms require further investigation.

While gene amplification in plants is often associated with genome duplication, the results obtained from gene duplication analysis indicated a relatively low frequency of this event in HvbHLH TFs. Consequently, it can be presumed that there are other factors at play influencing the amplification of HvbHLH TFs in barley.

BHLH TFs are known to fulfill crucial functions in various stages of plant growth and development (Guo et al., 2021). An examination of the expression patterns of HvbHLH in purple-grain Qingke and white-grain Qingke revealed the presence of a substantial quantity of HvbHLH TFs expressed at different developmental stages in Qingke. Notably, HvbHLH78 (also called HvAnt2) exhibits minimal or absent expression in the initial stage of both white and purple grains among the three stages. In contrast, its expression markedly increased during the late milk and early dough stages of purple grains in comparison to any of the stages in white grains (Fig. 5). The expression pattern of the HvAnt2 gene was similar to that of the gene responsible for encoding the key enzyme involved in anthocyanin synthesis (Yao et al., 2022). It was further determined that in Qingke, the HvANT2 gene is closely related to anthocyanin synthesis, which is consistent with previous findings (Cockram et al., 2010). Typically, the regulation of the flavonoid biosynthesis pathway takes place at the transcriptional level by the MYB-bHLH-WD40 TF complex (Li et al., 2022). This pathway gives rise to a range of flavonoid metabolites, including flavonols, anthocyanins, and proanthocyanidins (PAs). Therefore, we embarked on a series of studies focusing on HvAnt2 TFs to gain deeper insights into this intricate pathway.

Recent studies have identified a 179/168 bp insertion in the HvAnt2 promoter as the likely causative variant leading to white grains in barley (Dondup et al., 2023). The presence of purple grains in barley is caused by HvANT1 and HvANT2 (Xu et al., 2023). These studies indicated that HvANT2 was crucial for producing purple/white seeds in barley. However, further research is needed to study the specific structural genes associated with anthocyanin synthesis that are regulated by this gene. We conducted a Y1H study to determine which structural genes HvANT2 interacts with in the anthocyanin synthesis pathway. A known consensus sequence for the bHLH transcription factor has now been identified in maize (CANNTG) (Hartmann et al., 2005). In our study, by screening motifs that interact with the HvAnt2 protein, we were able to validate anthocyanin-related structural genes associated with these motifs. The results demonstrated that the HvAnt2 protein interacted with the structural genes HvGT, HvF3′H, and HvCHI within the anthocyanin biosynthesis pathway. Additionally, HvCHI and HvGT exhibit higher affinity for interacting with the HvAnt2 protein. The functions of GT, F3′H, and CHI in anthocyanin synthesis have been extensively characterized in previous research (Song et al., 2016; Zhang et al., 2020a; Han et al., 2012). Multiple studies have shown that HvANT2 interacts with the F3′H, ANS, CHS, CHI, F3H and DFR genes during anthocyanin synthesis, but interaction with the HvGT gene had not been reported (Chen et al., 2023; Gordeeva et al., 2019; Shoeva et al., 2016).

Conclusion

The HvAnt2 gene appeared to be of significant importance in anthocyanin biosynthesis, as evidenced by its distinct expression patterns in various tissues and its differential expression across different species. In this study, we identified 41 core motifs interacting with the HvANT2 protein using protein-motif interaction. Additionally, HvGT was a newly identified gene that interacted with HvANT2. This study not only underscores the pivotal role of HvAnt2 in anthocyanin biosynthesis, but it also provides a valuable reference for future investigations aimed at unraveling the regulatory mechanisms and functional aspects of anthocyanin biosynthesis in Qingke.

Supplemental Information

Supplemental Information 1 Conserved motif logo information of HvbHLHs

Supplemental Information 2 Promoter sequences of HvGT, HvCHI and HvF3′H

Supplemental Information 3 HvbHLH TFs identified in barley

Supplemental Information 4 A synthesis analysis was conducted comparing barley with Arabidopsis, rice, and maize

Supplemental Information 5 Raw data for qRT-PCR

Supplemental Information 6 Original images of interaction

We would like to thank LetPub for its linguistic assistance during the preparation of this manuscript.

Abbreviations

bHLH Basic helix-loop-helix

TF Transcription factor

qRT-PCR Quantitative reverse transcription polymerase chain reaction

CHI Chalcone isomerase

F3′H Dihydroflavonol-3′-hydrogenase

Y1H Yeast one-hybrid

GT UDP-glucosyltransferase

SD-TL Solid-deficient yeast media lacking both leucine (Leu) and tryptophan (Trp)

SD-TLH Solid defective yeast culture medium lacking tryptophan (Trp), leucine (Leu), and histidine (His)

SD-TLH + 50 mM 3AT Solid defective yeast culture medium lacking tryptophan (Trp), leucine (Leu), and histidine (His) combined with the competitive inhibitor 3-amino-1,2,4-triazole (3AT).

Additional Information and Declarations

Competing Interests

Author Contributions

Data Availability

The authors declare there are no competing interests.

Yan Wang conceived and designed the experiments, performed the experiments, analyzed the data, prepared figures and/or tables, authored or reviewed drafts of the article, and approved the final draft.

Lin Chen performed the experiments, analyzed the data, prepared figures and/or tables, and approved the final draft.

Youhua Yao analyzed the data, authored or reviewed drafts of the article, and approved the final draft.

Lupeng Chen performed the experiments, prepared figures and/or tables, and approved the final draft.

Yongmei Cui analyzed the data, authored or reviewed drafts of the article, and approved the final draft.

Likun An analyzed the data, authored or reviewed drafts of the article, and approved the final draft.

Xin Li analyzed the data, authored or reviewed drafts of the article, and approved the final draft.

Yixiong Bai analyzed the data, authored or reviewed drafts of the article, and approved the final draft.

Xiaohua Yao conceived and designed the experiments, prepared figures and/or tables, authored or reviewed drafts of the article, and approved the final draft.

Kunlun Wu conceived and designed the experiments, authored or reviewed drafts of the article, and approved the final draft.

The following information was supplied regarding data availability:

Transcriptomic data for Qingke at different developmental stages from our previous studies are available at Sequence Read Archive (SRA): SRR18355550, SRR18355551, SRR18355552, SRR18355554, SRR18355555, SRR18355556, SRR18355557, SRR18355558, SRR18355559, SRR18355560, SRR18355561, SRR18355562, SRR18355564, SRR18355565, SRR18355566, SRR18355567.

The raw qRT-PCR data are available in the Supplementary File.

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
