# Peer review of "Investigating the regulatory role of HvANT2 in anthocyanin biosynthesis through protein–motif interaction in Qingke"

_PeerJ, doi:10.7717/peerj.17736_

## Round 0.1 · original submission · Major Revisions

The manuscript got critical comments demanding major revision. Please consider the reviewers comments and suggestions.

It is important to cite references on anthocyanin biosynthesis in barley; extend the reference list.

The Ant2 gene was identified as the main regulatory gene of anthocyanin synthesis in barley tissues by GWAS in Cockram et al., 2010 DOI: 10.1073/pnas.1010179107, but this work was not mentioned in the manuscript.

The involvement of the gene in anthocyanin biosynthesis in grain was studied in (2016) DOI: 10.1371/journal.pone.0163782 and (2019). DOI: 10.1186/s12870-019-1638-9. These works were not mentioned in the text either.

The allelic diversity of Ant1 and Ant2 genes related to grain color in Tibet barleys were studied in Dondup et al., 2023 DOI. 10.3389/fpls.2023.1189642, that also was not mentioned in the manuscript. No one manuscript focusing on Ant2 gene as anthocyanin biosynthesis regulatory gene was mentioned.

Reviewer 1 ·

Basic reporting

The manuscript “Identification of Hordeum vulgare basic helix-loop 2 helix genes reveals candidate genes related to anthocyanin biosynthesis in Qingke barley” considers the bHLH gene family in barley with focus on Ant2 gene, that control anthocyanin biosynthesis in barley plant tissues, including grain.
The MS have a merit, in my point of view, in the part of identification all gene members of the big family based on new barley genome assembly data, but there is no novelty in identification of Ant2 gene as a main regulator of anthocyanin biosynthesis.
The gene was identified as main regulatory of anthocyanin synthesis in barley tissues by GWAS in 2010 and was studied in respect to grain pigmentation in 2016 and 2019, but none of the primary sources were cited in the manuscript. Instead that the authors stated that “Through RNA-Seq of purple and white grain Qingke, we discovered a significant transcription factor, ANT2 (HvbHLH78), associated with anthocyanin biosynthesis.” And “These findings provide substantial evidence for the pivotal role of ANT2 TF in anthocyanin biosynthesis” . This do not correspond to the reality.
I recommend revising the manuscript substantially and do not make a focus of the manuscript on already known data. As a novelty I can recommend to highlight the data on motifs found and confirmed by Y1H assay.

Experimental design

Y1H assay was not explained properly. What is meant by SD-TL, SD-TLH, and SD-TLH+50 mM 3AT and how does the test work?

In lines 148-152 the procedure is not clearly explained. Were the vectors commercial and who produced them? As I remember the Y1H assay is used to study protein-protein interaction but here as I understand the authors study interaction of regulatory protein and cis-elements of promoter. Please, describe the procedure properly or cite the proper article, where it described.
Figure 6 – What are abbreviations SD-TL, SD-TLH, and SD-TLH+50 mM 3AT and how does the test work?

Line 242 “Through sequencing comparison….”. Please clarify in material and methods how did you find the motifs, there is no information at all about the approach used to find them. What gene sequences were used for this analysis, add the table to Supplementary materials. Table 1 is not informative .

How were the primers constructed for expression anlaysis? Describe the design procedure or cite the proper article, in which they published.

Validity of the findings

no comments

Additional comments

The authors found conserved motifs in the bHLH genes, but they did not show which of them comprise bHLH one. This information should be added in Figure 3, and the motifs identified should be analyzed and discussed additionally if possible.
In addition there are minor points.
In Lines 125-130, the information correspond to RT-PCR but not RNA-seq analysis. RNA-seq data expressed as FPKM but not fold-changes. Moreover , lines 127-130 are identical to 138-141.

Line 138. What you mean under the phrase “The RT-qPCR reaction system was performed by Yao et al…”?
Please pay attention, in the text both versions of the name of gene Ant2 and HvAnt2 are found, please, make uniformly throughout the whole text.

There are repits in references: references 4 and 7 are identical, as well as 1 and 12, and 17 an 20. Please, carefully check the references on duplicates.

Line 310 Cite the proper work. There is no citation in reference list on Chen et al.
Line 340-341, and 344-347 – there is no novelty here. These data have been already published in scientific literature. Re-write the discussion and focus in novelty of the study but not ignore discoveries that have been already made and published open access.

·

Basic reporting

In this manuscript entitled "Identification of Hordeum vulgare basic helix-loophelix genes reveals candidate genes related to anthocyanin biosynthesis in Qingke barley (#96506)" by Wang et al., through transcriptomic analysis, they identified 161 bHLH genes and the structural genes related to anthocyanins interacting with HvANT2 were verified by using a yeast one-hybrid (Y1H) assay.
The work in this manuscript is logical and clearly presented. However, I still have some comments and would recommend the work for publication after major revise.

The comments and suggestions are listed as follows.
1. The Abstract is too long. It needs to be condensed.
2. The growth status of the plant needs to be defined in more detail.
3. The discussion section should be re-organized based on the results in the current study.
4. The language needs to be polished and improved,
5. Overall, there is a lot of text in the picture, resulting in a lack of clarity
6. Lines 442-443,The "L" in "(Setaria italica L.)" should be not italic.
7. It is suggested to increase the latest relevant literature appropriately and reduce some irrelevant literature.

Experimental design

.

Validity of the findings

.

Additional comments

.

---

## Round 0.2 · Minor Revisions

The manuscript became better after the revision. But, as the reviewer noted, some questions are still remained. The main concerns is the experimental verification of the predicted interaction. This part should be described in more details. computational experiment could be considered. Then change the conclusion, present it as hypothesis of protein-motif interaction. Please revise accordingly.

Reviewer 1 ·

Basic reporting

The authors have revised the manuscript while including all the comments rose during the last review process. However some questions are still remained.

Experimental design

no comments

Validity of the findings

The major question is the interaction of the motifs identified and ANT2 transcription factor.
The Y1H assay has not been clarified properly. Why in figure 6, the yeast colony marked as negative control grows on SD-TLH medium. If it is a negative control, then the interaction between motif and ANT2 regulatory factor is absence and the growth on "–His" medium would be inhibited. As describe in manual for pHIS2 vector, the gene HIS3 is required for the histidin biosynthesis that in turn is activated when interaction between motif and the transcription factor occurs. So, it is expected to see absence of growth in negative control in –his medium, but here we see the colony growth of negative control on –his medium. One can conclude that the assay works not properly. In the legend to the figure, it is not described what positive and negative control is. Is it the same as in Figure 7 legend? Could you explain what mean the growth of colony on each medium and when 3AT was added?
In the figure 7, the negative control pGADT7-ANT2+pHIS2-p53 looks the same as the tested motifs. It means that there is no interaction between ANT2 and these motifs. Only for the F3’h some differences in yeast growth are observed between the tested construction and negative control pGADT7-ANT2+pHIS2-p53.

Additional comments

1. In the sentence *Motifs that interacted with pGADT7-HvANT2 were identified by gene detection, followed by DNA sequencing and a comparative analysis of the positive clones*, it is not clear how the motifs were identified by gene detection.
2. In the sentence, *eventually, we identified three enzymes encoding anthocyanin structural genes, namely, chalcone isomerase (CHI), dihydroflavonol-3'-hydrogenase (F3'H), and UDP-glucosyltransferase (GT)* explain, please, how you identified these genes, why you did not identify the genes Chs, F3h, F3’5’h, Dfr, Ans, that also participate in anthocyanin biosynthesis.
3. There is a known consensus sequence for bHLH transcription factors – CANNTG, that was not discuss in the manuscript (Hartmann U., Sagasser M., Mehrtens F., Stracke R., Weisshaar B. Differential combinatorial interactions of cis-acting elements recognized by R2R3-MYB, BZIP, and BHLH factors control light-responsive and tissue-specific activation of phenylpropanoid biosynthesis genes. Plant Mol. Biol. 2005. Vol. 57. P. 155-171)

·

Basic reporting

The revised manuscript has much improved. All previous questions and suggestions have been addressed.

Experimental design

no comment

Validity of the findings

no comment

Additional comments

no comment

---

## Round 0.3 · accepted · Accept

The reviewers have no more remarks. The manuscript shall be accepted.

Reviewer 1 ·

Basic reporting

All coments were addressed by authors of the manuscript.

Experimental design

It was evaluated in previous round of the reviewing process.

Validity of the findings

The additional data were added to the supplementary materials which prove the validity of the findings.

·

Basic reporting

The revised manuscript has much improved. All previous questions and suggestions have been addressed.

Experimental design

no comment

Validity of the findings

no comment

Additional comments

no comment